# Cutting Behavior of Al_0.6_CoCrFeNi High Entropy Alloy

**DOI:** 10.3390/ma13184181

**Published:** 2020-09-20

**Authors:** George Constantin, Emilia Balan, Ionelia Voiculescu, Victor Geanta, Valentin Craciun

**Affiliations:** 1Robots and Manufacturing Systems Department, University Politehnica of Bucharest, 060042 Bucharest, Romania; george.constantin@icmas.eu (G.C.); emilia.balan59@yahoo.com (E.B.); 2Quality Engineering and Industrial Technologies Department, University Politehnica of Bucharest, 060042 Bucharest, Romania; 3Engineering and Management of Metallic Material Processing Department, University Politehnica of Bucharest, 060042 Bucharest, Romania; victorgeanta@yahoo.com; 4National Institute for Laser, Plasma and Radiation Physic, 077125 Magurele, Romania; valentin.craciun@inflpr.ro; 5Extreme Light Infrastructure-Nuclear Physics, IFIN-HH, 077125 Magurele, Romania

**Keywords:** high entropy alloy, milling, machinability, cutting, microhardness, roughness

## Abstract

There is an increased interest in high entropy alloys as a result of the special possibilities of improving the mechanical, physical or chemical characteristics resulting from metallic matrices made of different chemical elements added in equimolar proportions. The next step in developing new alloys is to determine the cutting conditions to optimize manufacturing prescriptions. This article presents a series of tests performed to estimate the machining behavior of the Al_0.6_CoCrFeNi high entropy alloy. The effects of temperature during machining, wear effects on the cutting tool, evolution of the hardness on the processed areas, cutting force components and resultant cutting force for high entropy alloy (HEA) in comparison with 304 stainless steel, scrap aspect and machined surface quality were analyzed to have an image of the HEA machinability. In terms of cutting forces, the behavior of the HEA was found to be about 59% better than that of stainless steel. XRD analysis demonstrated that the patterns are very similar for as-cast and machined surfaces. The wear effects that appear on the cutting edge faces for the tool made of rapid steel compared to carbide during HEA machining led to the conclusion that physical vapor deposition (PVD)-coated carbide inserts are suitable for the cutting of HEAs.

## 1. Introduction

Alloys design concept was first based on the idea of a “base element”. In this regard, one or at most two basic elements are considered to form the metallic matrix and other chemical elements, added in different minor proportions, have the role to enhance specific properties for certain applications [1,2,3]. In contrast to classical methods, a new concept of materials based on the mixing of five or more different elements for the obtaining of so-called high entropy alloys (HEA) emerged. These recently discovered advanced materials [4] have been in the attention of researchers all over the world, both academics and technologists [5,6]. HEAs became important due to the uniqueness of their composition, simple and micro-amorphous microstructure and their functional properties [7,8,9].

To obtain the high entropy into the materials, five or more main elements must be mixed in similar proportions, comprised between 5 and 35 at. %. The most commonly used metallic elements are Al, Cr, Fe, Co, Ni, Cu, Ti, etc., but nowadays other refractory elements are added to form very hard metallic matrices (Ta, Zr, Hf and W) [10,11,12]. HEA materials are often considered to be solid nanostructured alloy solutions without identifying the solvent and solutes. Therefore, the current theories of solid solutions are difficult to be applied for identifying mechanisms for obtaining HEAs [13,14,15]. More than 300 HEAs with different microstructure have been studied so far, by combining of more than 30 different elements [1,16,17]. These alloys have been highlighted by special functional properties, such as high hardness, durability, high heat resistance and corrosion resistance. More than 400 research papers have been published to date focusing on the composition and structure influence on the material behavior [5]. However, the understanding of all aspects specific to the HEA material production and processing is still in the early stage [6,10,16]. HEAs possess other properties as well, such as a favorable combination of resistance to compression and ductility, distinct electrical and magnetic properties and biochemical properties [14,18]. In most research, HEAs are characterized in detail by mechanical tests concerning hardness, tensile, compression, fracture toughness, creep and fatigue [19]. The composition and structure of AlCoCrFeNi type materials determine their mechanical properties. Thus, the hardness of these materials increases with the Al content, depending on the microstructure FCC, BCC + FCC and BCC phases [4,20]. The Vickers hardness values vary as follows: FCC, 100‒200 HV; BCC + FCC, >600 HV; and BCC, from minimum to maximum values with increasing BCC content. The hardness and strength are influenced by some determinant factors [16], such as each phase hardness or strength, relative volume of each phase, component phase morphology and distribution.

HEAs can be used as materials for engines (including aircraft engines), nuclear power stations, chemical plants, marine structures, tools, nozzles, stamping presses, molds, hardened surfaces, functional coatings, refractory constructions and coatings for electronics [1,5,21]. Other fields of applications could be the electronic, electrical, thermoelectric, electromagnetic and medical ones [16]. The most studied mechanical properties of the HEA alloys are the hardness and compression that allow them to be used in military applications [22,23]. Consideration should also be given to the cost and ease of manufacturing of HEAs, characteristics that have not been widely treated so far. It is perhaps time to initiate concentrated efforts to develop special HEA alloys for specific uses [7]. Researchers in the field of new metallic materials consider that HEAs and other similar alloys (e.g., high entropy ceramic (HEC) polymers) could solve many bottlenecks encountered in conventional materials, and that they will find their applicability in areas much broader than those listed above [1,16]. To extend the fields of application of these new materials and to highlight the link between the HEAs’ microstructure and their properties, new methods of characterization and processing must be defined and proposed, accompanied by laborious experimental research and advanced assisted data processing methods [15]. In addition, the changes in the concentrations of alloy components result in a very high diversity of possible alloys with specific properties. Thus, this opens wide opportunities for researching new properties of these materials.

Material machinability is very important for the technological stage and process planning. In the case of new materials, characterized by completely new chemical composition and different mechanical properties, machinability is the main criterion doe material choice [24]. Many aspects of HEA cutting should be studied in the framework of machinability. Both the working conditions and the mechanical and physical properties of the material are very important in characterizing machinability. The machinability is determined by several influence factors [25,26] that are divided into input and output factors. The input factors are independent of the cutting process and are the following: characteristics of the material to be machined, cutting material of the cutting tool and its body, tool geometry, parameters of the tool’s engagement in the workpiece, cutting conditions, type of processing, lubrication, part holding and rigidity of the machine tool. The output factors are dependent on the cutting process and include cutting forces, chip shape, process temperature, tool wear and life, surface quality, dimensional surface accuracy and specific energy consumption.

Most of the research on the machinability investigates the effect of the cutting parameters on an output parameter, such as roughness of the processed surface. Other research considers the combined effect of the cutting parameters on an output parameter [27,28]. A criterion may often have an influence contrary to another one. Therefore, for a larger and more precise study that gives an image of the behavior of a material, the multi-criterion assessment leads to a general approach [29,30].

Recent research addresses various aspects of alloy machinability, discussing issues involving mathematical modeling, numerical simulation and multi-criteria optimization. The work in [31] presents notable achievements regarding intelligent optimization of process parameters in turning of hardened high-carbon steel AISI 1060 using evolutionary algorithms to eliminate wastage of materials and energy. Roughness and surface topology are investigated based on optimization of parameters such as cutting speed and feed rate to minimize surface roughness in turning of a titanium alloy (Ti-6Al-4V) [32] and machining of CoCrMo alloy [33], or using a mathematical model of the relative position between cutting tool and machined surface in cutting of low carbon alloy hardened steel [34]. Zha et al. [35] brought important contributions on improving the cutting efficiency in the processing of nickel-based alloys using the finite element method and analysis of cutting forces and temperature.

There are few published research papers dealing with aspects of the HEA-type materials machinability. Polishetty’s work treats only the specific aspects, such as the forces in the cutting process and the roughness of the achieved surfaces, detaching the advantages and disadvantages of the HEA’s milling [36]. In Guo’s work, several machining processes applied to HEAs obtained through laser additive manufacturing, such as milling, grinding, mechanical polishing, electrical discharge machining and electro-polishing, are treated in turn [37].

The processes are evaluated considering the surface morphology, surface roughness, elemental composition, microhardness, residual stress and surface quality, along with some conclusions regarding the use of machining in HEAs. Compared to almost all studies on new materials in terms of mechanical and metallurgical properties, such as microstructure, phase constitution and tensile properties, this article proposes the study of the cutting machinability of HEA materials. For this purpose, the temperature during machining, wear effects on the cutting tool, evolution of the hardness on the processed areas and the cutting force components measured during machining of HEA and derived resultant cutting force were analyzed in comparison with 304 stainless steel. In addition, the chip appearance and surface quality (roughness and aspect of the machined surface) along with microstructure and surface changes during machining were examined. All these studied aspects give a complex view of the HEA processing by cutting, proving the importance and novelty of the research that contributes to a better understanding of the behavior of machined HEA and to the shortening of the implementation period of HEA materials in industrial applications.

## 2. Experimental Procedure

### 2.1. Experimental Setup for Milling Tests

The cutting tests were achieved by using a shank cutter produced by Pokolm with a diameter d_1_ = 12 mm and two indexable inserts for high feed milling having carbide grade HSC 05 and coating PVTi. The cutting tool was coupled to a thermal contraction HSK 50/BT 40 tool holder. The main dimensions of the milling cutter (Figure 1a) and insert (Figure 1b) used were: length l = 6.2 mm, radius r = 20 mm, clearance angle α = 11° and entering angle κ = 93°. The rack angle γ of the rack faces of the inserts mounted in the tool body was kept zero.

The experimental study was carried out in dry cutting conditions on FIRST MCV 300 vertical milling machining center having three NC axes equipped with Siemens Sinumerik 840D Numerical Control that belongs to the Machine Tools Laboratory of Robots and Manufacturing Systems Department from the University Politehnica of Bucharest. The main working parameters are: maximum spindle speed of 8000 rpm, feed rate of 10^3^ mm/min and 7.5 kW drive motor. The experimental setup for cutting force component measuring is shown in Figure 2 and contains the following components: cutting tool fixed in the main spindle and sample mounted on the table of the dynamometer Kistler 9257B fixed on the table of the vertical machining center First MCV 300. The signal issued from the dynamometer is processed by the multi-channel signal amplifier of 5070A type that passes it to an acquisition board of PCIM-DAS1602/16 type. The computer uses acquisition and processing software DynoWare Type 2825A. The signals are processed and recorded in specific files (.dwd) and can be visualized on the computer screen. These files can be transformed into text files for importing them in specific programs (Excel, Matlab, etc.) in order to be processed.

### 2.2. Materials and Methods

During the experimental research, the cutting tests were achieved on two materials, namely Al_0.6_CoCrFeNi alloy (HEA) and stainless steel X5CrNi18-10 (304 after AISI) [38], having some similarities from the point of view of the chemical composition (only 0.06 wt.% C), toughness, and hardness. The experimental Al_0.6_CoCrFeNi alloy has the chemical composition shown in Table 1. The code for the Al_0.6_CrFeCoNi alloy corresponds to aluminum atomic ratio x = 0.6. This alloy was made of technical pure elements, as follows: Cr 99 wt.%; Al 98.5 wt.%; Co 99.5 wt.%; Ni 99.5 wt.% and MK 3 extra ductile steel, having 98.5 wt.% Fe. The metallic materials were melted in electric arc under argon-protective atmosphere in MRF-ABJ 900 equipment, at ERAMET Laboratory from University Politehnica of Bucharest, using a non-consumable tungsten electrode. The homogeneity of the samples was ensured by remelting them five times on each side.

Spectrometric tests were used for determining the chemical composition, within the Laboratory LISEOFRX, Research Centre in Biomaterials, University Politehnica of Bucharest, using a spark optical emission SPECTROMAXx spectrometer. The microstructure of the samples was investigated using Optical microscope Olympus RX51 equipped with Analysis software for images processing, Inspect S SEM microscope equipped with Z2e EDAX sensor (LAMET Laboratory from University Politehnica of Bucharest) and X-ray diffraction with an Empyrean diffractometer working with a Cu anode at 45 kV and 40 mA power (National Institute for Laser, Plasma and Radiation Physics, Magurele, Romania). The X-ray diffraction instrument was set in parallel beam geometry with the aid of a mirror on the incident beam side to allow for grazing incidence XRD investigation. Microhardness Shimadzu HMV 2T (Shimadzu Corporation, Tokyo, Japan) was used for performing hardness tests. The quality of the surface processed by milling was assessed using roughness tests performed with the Mitutoyo SJ-210 portable surface roughness tester (Mitutoyo, Huston, TX, USA) with diamond tip that can perform differential induction measurements within a range of 350 µm (−200 µm; +150 µm).

### 2.3. Sample Preparation for Cutting Tests

The HEA samples were provided in a cast state, with irregular shape. The visual control pointed out their irregular geometry, some porosity and visible deformations. The hardness tests carried out on the cross sections and the exterior surfaces showed that, at level of the surface layer, the material was hardened during the cooling process after casting. Therefore, the sample surfaces for cutting tests were prepared to remove the hardened surface layer, obtain known dimensions, plane surfaces needed for holding the sample on the dynamometer table and keep a constant cutting depth during milling.

The challenge concerning the cutting parameters to be used in machining on the conventional milling machine appeared from the beginning. Some initial cuts were carried out for calibrating the processing speed and feed rate, due to the intense heating and high sound (vibrations) generated during cutting.

After the initial tests, the working parameters for machining were set. As a precaution measure, greasing was applied in the cutting area. The samples were fixed on the dynamometer table during machining to have a plane horizontal initial surface. The samples were machined at the dimensions 145 × 15 × 6 mm^3^ for HEA and 150 × 12 × 20 mm^3^ for stainless steel.

### 2.4. Experimental Plan Used for Milling Tests

A good experimental plan leads to a minimum number of experiments and also the highest accuracy of results. Therefore, for researching the behavior of HEA and of the reference material 304 stainless steel during the cutting process by milling, the Taguchi method having three parameters was chosen. The face milling parameters or independent variables are the cutting speed v_c_ (m/min), feed per tooth f_z_ (mm/tooth) and axial depth of cut a_p_ (mm). For these parameters, five values or levels were considered (Table 2). Having the set of parameters, the orthogonal matrix chosen resulted in an experimental plan of 25 tests (Table 3).

The radial depth of cut a_e_ was kept at 100%, exactly a_e_ = 12 mm (machining with full tool diameter). The machining tests were carried out by dry cutting. This experimental plan is useful also for statistical data processing, including generating models of the studied output parameters valid over the ranges proposed and with possibility of extending the ranges [39].

## 3. Results

This section provides an accurate description of the experimental results obtained and their interpretation. The aspects regarding temperature during machining, cutting tool wear, microhardness of the processed surfaces, cutting forces, chip analysis, microstructure, chemical composition of the machined surface and machined surface quality are treated, respectively.

### 3.1. Study of Cutting Temperature in HEA Milling

The cutting temperatures during the HEA cutting were acquired by indirect measurement using a thermo vision camera Therma CAM SC 640 (FLIR Systems, Boston, MA, USA) equipped with Therma Cam Researcher Pro 28 SR-2 software. The measuring range of the apparatus is of 0–500 °C, the sensitivity of the camera is 0.006 °C, measurement accuracy is about ±1 of the measured value and the measurements were achieved using a pre-set emissivity of 0.8. The temperature measurements were made during the cutting process in the conditions established through the cutting measurement schedule simultaneously with the measurements of the cutting forces. During cutting process, the camera visualizes the cutting tool body, inserts, sample and detached chips (Figure 3a). A closer position of the camera and thermocouple technique used for orthogonal cutting [40] is not suitable during milling process. The measured temperatures are relative temperatures and cannot be considered as those of the insert–chip interface. The experimental data taken by the thermo vision camera are processed by the associated program. The maximum temperature is recorded on the surfaces of the detached chips close to the cutting area. From the processing area, a significant rectangle is selected together with a temperature line crossing it, along which the temperature diagram is represented. An example of the most significant temperature evolution during cutting process is shown in Figure 3b.

The maximum value of temperature reached after about 6 s of processing is T_max_ = 213.2 °C. One can distinguish between the diagram parts: Tool engaging in material where the temperature is increasing; full diameter cutting, in which the maximum temperature tends to stabilize, however, with a slight increasing trend; and tool exit of cutting with the attenuation of the peaks.

As the diagram of the temperature variation in the cutting area for the successive 25 tests shows (Figure 4), the temperature variation range is 120‒281 °C, similar to the values reached in the case of stainless steel. The maximum temperature values (240‒281 °C) were obtained for Tests 5, 9, 13, 20, 21 and 25 (Figure 4), characterized by the highest depth of cut value, a_p_ = 0.42 mm. In a similar approach, the smallest temperature value was obtained for the minimum value of the axial depth of cut, a_p_ = 0.1 mm. From this point of view, the main influencing parameter of the cutting temperature is the axial depth of cut.

### 3.2. Tool Wear during HEA Cutting

The cutting experiments were carried out using a new insert having two cutting edges. Cutting Edge 1 (CE1) was used in relatively hard conditions, for removing the top layer of the cast material, in order to obtain a plane surface ready for the cutting tests. Therefore, for removing the very hard and inhomogeneous layer, the parameters used were: speed n = 650 rpm, feed f = 0.05 mm/tooth, feed rate v_f_ = 65 mm/min and axial depth of cut a_p_ = 0.2 mm. This first step was carried out to check the parameter values in order not to affect the tool. The indications from the tool producer refer only to a homogenous material, similar to stainless steels.

The machining process was continued using the second set of parameters: n = 1000 rpm, f_z_ = 0.05 mm/tooth, v_f_ = 100 mm/min and a_p_ = 0.2 mm. The total length of cut was l_cut_ = 3000 mm (machining time t_m_ = 30 min). After processing the sample, the cutting edge wear was visible on the rack face and flank of CE1 (Figure 5). The wear on the rack face (Figure 5a) is highlighted by the maximum crater width KB_max_ = 378 µm and crater front distance KM_max_ = 166 µm. One can see the affected area, where the hard layer of the insert material has been removed. A crater is observed, being defined by KB = 121 µm and KM = 84 µm (Figure 5b). Due to its ease of measurement, flank wear is the most commonly used. The flank wear land and maximum flank wear width VB_B max_ = 86 µm are highlighted (Figure 5c). A detail of the most affected area of the flank wear land is shown in Figure 5c.

The next stage was the material processing under the cutting conditions of the 25 tests. This was carried out using Cutting Edge 2 (CE2). In this case, the total length of cut was of l_cut_ = 300 mm (machining time t_m_ = 10 min). The cutting edge wear on the rack face and flank of CE2 is shown in Figure 6. The wear on the rack face (Figure 6a) is defined by the maximum altered area width KB_max_ = 490 µm and crater front distance KM_max_ = 225 µm. One can see the affected area by removing of the coating layer and the of basic insert material. A small crater is observed, being defined by KB = 134 µm and KM = 83 µm (Figure 6b). The flank wear land and the maximum flank wear width VB_B max_ = 67 µm were highlighted on the flank (Figure 6c). A detail of the most affected area of the flank wear land is shown in Figure 6c. The biggest one presents larger erosion in volume and depth (Figure 5b) than the craters on the rack face of the CE2 (Figure 6a,b). As regards the wear on the flank, CE1 (Figure 5d) is slightly more worn than that of CE2 (Figure 6d).

Inserts for precision machining are often subjected to nitriding process to increase hardness and wear resistance. This process allows increasing the machining performance due to WN compound and protective layer formation [41]. As can be seen in Figure 7, on the cutting edge wear area, the whole protective nitrided layer was removed by the erosion effect of metallic scraps.

### 3.3. Microhardness of the Al_x_CoCrFeNi Alloy Processed by Milling

Self-hardening of machined surfaces during cutting shows interest for investigation for the new HEA. This phenomenon can be revealed by studying the hardness of the machined surfaces in the case of cutting by milling. The microhardness was measured on each surface processed setting a test force of 0.2 N and an application time of 15 s, at an environmental temperature of 25 °C and humidity of 55%. To obtain acceptable results, five determinations for each area corresponding to a cutting test of the experimental schedule were made, taking into account the hardness average value. Some values deviating slightly from the trend are observed (Figure 8).

### 3.4. Cutting Forces in Milling of HEA

From the point of view of a definition of machinability, this is usually assessed having as reference the material 160 Brinell AISI B 1112 free machining low carbon steel [26]. However, the machinability of the new HEA considering the cutting forces criterion was analyzed in comparison with a metallic material having similar hardness and toughness. According to the recommendations given by metallurgical specialists and also by cutting tools producers, 18Cr10Ni stainless-steel alloy (E 304) was considered as reference for comparison. It was considered as having the reference machinability rating of 100%. Alloys having a rating less than 100% are considered more difficult to machine in opposition with those easier to machine and having a rating above 100%. In the paper, the average values of the cutting force are considered. After performing all the 25 machining tests for the two materials (HEA and 304 stainless-steel), the data were recorded and processed. Three charts were considered for comparing the feed force evolution of the two materials (Figure 9).

Usually, as a machinability criterion, the cutting force is defined by the two components: cutting force and feed force. In this case study, the cutting force is associated with the component F_y_ and the feed force with F_x_. In addition, the axial component F_z_ was analyzed. The dynamometer reference O_m_ X_m_ Y_m_ Z_m_ system was aligned with the machine tool OXYZ system (see Figure 2) and the measured values of the forces were considered with their signs given by the direction along axes (+ or −).

According to Measuring Points 5, 9, 13, 17, 21 and 25, the differences between the two diagrams reach the maximum values. The measurement points are characterized by Level 5 of the depth of cut a_p_. For these points, the machinability rating ranges between 150–200% for F_y_ and 150–160% for F_z_. The approach of cutting force components on axes can be only a partial and simplified concept. In fact, in the cutting process, the resultant cutting force acts. The real cutting force expression is based on the axial components:(1)$$F_{\mathrm{cut}}=\sqrt{F_{x}^{2}+F_{y}^{2}+F_{z}^{2}}$$

Based on Equation (1), the values of the machining rating during HEA processing by milling, taking in consideration as reference the stainless-steel E 304, were calculated (Figure 10).

### 3.5. Chip Analysis

To estimate the behavior of the HEA alloy in machining, fragments of chips resulting from cutting were examined by scanning electron microscopy. The chips were selected from Experiment 13 for both HEA alloy and E 304 stainless-steel as a reference. The macroscopic appearance of the metallic chips reveals that both materials form small corrugations of shorter lengths of 2‒4 mm, with sharp edges and striated surfaces on the inner zones (Figure 11a,d). It is noted that, in both analyzed cases, the striations formed on the chip surface result from fragmentation and slipping of the deformation plane in the material (Figure 11b,e). In the case of stainless-steel, a higher plasticity of the material is evidenced by the formation of quasi-symmetrical deformation planes (Figure 11f), while the breaking and sliding planes are irregular in HEA (Figure 11c). This behavior may be due to the higher inhomogeneity of HEA compared to high-grade stainless-steel.

Regarding the chemical analysis of HEA chips, small adherent zones were formed on their outer surface containing different concentrations of the chemical elements present in the base alloy (Figure 11b). In addition, in this case, oxides are located on the outer face of the chip as alternant layers having different chemical compositions.

The large percentages of chemical elements, such as C (42 wt.% C), Ca (13 wt.%) and Si (11 wt.%), can be due to adherent deposits on the chips and probably come from the working environment (tools or handling by operators). On the chip surface, the oxidation effect is important (69 wt.% O) and similar to those observed on the 304 chip. Some of the adherent areas also contain larger amounts of aluminum coming from HEA (Table 4 and Figure 12). The analysis of the local chemical composition on the impurity fragments observed on the internal surface of the stainless-steel chip revealed oxidation and segregation processes of chemical elements such as Cr, Ni and Mo (Figure 11e).

Thus, concentrations of over 35 wt.% Cr and between 9 wt.% Ni and 13 wt.% Ni were measured on the striation flanks (Table 5 and Figure 13).

Significant values of oxygen content (60 wt.% O) and large differences in Cr, Fe and Al contents in different measurement areas on the analyzed fragment indicate that strong oxidation effects are produced during material processing.

### 3.6. Machined Surface Quality

The roughness development over the 25 tests reveals the roughness range of 0.39–2.058 µm (Figure 14). This development is usual in most common metallic materials, including stainless-steel.

The best roughness values were obtained for Tests 6, 11 and 16, where the feed is minimum, f_z_ = 0.05 mm/tooth, as well as for 17 (f_z_ = 0.07 mm/tooth) and 18 (f_z_ = 0.09 mm/tooth). For the last two tests, these small deviations of the measurements can be caused by the imperfections of the alloy structure.

The machining surfaces in the 25 tests were analyzed using an optical microscope. Figure 15a shows the surface processed for Test 4 and Figure 15b the surface obtained in Test 8. The obtained images highlight the traces left by the cutting edges of the tool specific to the two inserts and also the appearance of the surface processed. Thus, the surface macro-geometry, which corresponds to the roughness produced by the cutting process, reveals the occurrence of surface defects caused by the material pulling out in the cutting process as well as those caused by the imperfections of the alloy structure (cracks, craters and voids).

The highest roughness values are recorded for Tests 5, 10, 15, 20 and 25, where the feed is the highest (f_z_ = 0.15 mm/tooth) and for Tests 1–4, in which the deviations from the expected roughness occur with an offset of approximately 1 µm with regard to the rest of the diagram. It is the same situation as for Test 5, in which the maximum value of the parameter is recorded.

On HEA machined surface of Test 25, chemical composition measurements on micro-area by EDAX method indicate a low oxidation effect, with a concentration of about 0.58 wt.% O (Figure 16).

### 3.7. Microstructure Analysis

For performing the microstructure analysis, the surfaces of the samples were embedded in phenolic resin, and then polished using SiC abrasive paper (grit 360–600) and alpha alumina powder. Then, they were electrochemically etched in a solution of 100 mL of water and 10 g of oxalic acid for 10 s. Prior to analysis, the sample surface was ultrasonically cleaned for 10 min in propanol and then dried in hot air. The microstructure of as-cast HEA is dendritic (Figure 17a). In the dendritic area, elements such as Co, Cr, Ni and Fe are evenly distributed, while the interdendritic areas are rich in Al (Figure 17c). The chemical composition measured by EDS analysis is very similar with those determined by spectral analysis (Table 1).

In the case of machined alloy, analysis performed near the machined surface indicates the same dendritic aspect of the microstructure, but also some changes in chemical composition (Figure 18). There, the content of Al and Ni is higher compared to the bulk area, which allows the formation of different compounds containing Al, Ni and Fe located interdendritic.

To evaluate the structural changes produced by cutting, the alloy phases were identified both for the cast material and for the processed Surface 25 (corresponding to the highest value of the cutting force). The structure of the base material Al_0.6_CoCrFeNi was analyzed by GIXRD using symmetrical theta–2theta geometry. The structure of the machined Region 25 was analyzed by grazing incidence X-ray diffraction (GIXRD) using incidence angles of 0.5° (penetration depth of around 0.106 mm), 2° (penetration depth 0.430 mm) and the symmetrical theta–2theta geometry (penetration depth from 3 to 10 mm).

The recorded patterns are displayed in Figure 19, where * denotes powder diffraction file 04-022-2301, Co0.25Cr0.25Fe0.25Ni0.25, space group Fm-3m and *a* = 3.58 Å, and # denotes powder diffraction file 004-018-5047, Al0.4Co0.4Cr0.4Fe0.4Ni0.4, space group Pm-3m and *a* = 2.8780 Å. Similar good matches are Al0.3Cr0.3Fe0.4, PDF 04-019-3848, space group Im-3m and lattice parameter *a* = 2.9160 Å and Co0.48Fe0.33Ni0.19, PDF 04-016-6385, space group Fm-3m and lattice parameter *a* = 3.5480 Å. Other low symmetry phases might be present in very small amounts.

The identification of phases is displayed in Table 6. It should be mentioned that HEA are quite new materials and the number of reference XRD patterns available in the database is still limited.

## 4. Discussion

### 4.1. Temperature Evolution during Machining

The temperature variation during the machining tests was in the range of 120–281 °C, having values similar to those reached for stainless-steel machining. There is a direct influence of the depth of cut on the cutting temperature. For bigger values of the cutting parameters used in rough processing, it is recommended to use cooling and lubrication. Preliminary cutting tests showed that conventional cutting tool materials, such as rapid steel, have very limited life and suffer an abrasion wear process resulting in a large amount of heat in the cutting process. Based on these observations, it has been concluded that conventional rapid steel cannot be used for HEA cutting. A good behavior during HEA machining has been provided by the inserts made by carbide grade HSC 05 PVTi coated, which can be used in good conditions without coolant.

### 4.2. Cutting Edge Wear

The machinability study was performed in two stages. In the first stage, a superficial processing of the entire specimen was performed, in order to remove the oxidation layer resulting from casting. Analyzing in detail the surface of the active area of the insert, it was found that the wear effects generated during the primary processing were more intense.

The analysis of the worn surfaces of the insert highlighted the tendency of progressive exfoliation of the protective layer of Ti deposited by the PVD method and the formation of craters on the rack face. The biggest worn area observed on cutting edge (CE1) rack face has been defined by dimensions of a crater (KB = 121 µm and KM = 84 µm). When checking wear area, small deposits were found on the rack faces that do not affect the tool geometry. Based on the findings, it can be concluded that the sample is made of a cast and inhomogeneous material, which has a variable hardness on the superficial layer. Some differences between the behaviors of the two cutting edges given by the sequence of engagement in the hardened surface of the material have appeared.

The CE2 involved the machining tests shows a larger affected area on the rack face presenting loss of coating layer and relatively reduced basic material volume (KB_max_ = 490 µm and KM_max_ = µm), this being caused by the wider range of the depth of cut of 0.1‒0.42 mm, compared to that of 0.2 mm used in the CE1. The larger crater is characterized by KB = 134 µm and KM = 83 µm having a smaller deep than that of CE1. The wear on flank indicates that wear of CE1 is more intense than that of CE2 (VB_B max_ = 86 µm compared to VB_B max_ = 67 µm). Even if wear appeared on the working zones, the required tool geometric parameters were kept during the cutting tests.

During HEA processing, some irregular small metallic deposits rich in Al (12 wt.%), Co (3.29 wt.%), and Cr (1.18 wt.%) were randomly deposited on the cutting edge. They influence the heat transfer, cutting edge geometry, and chip detaching mode. Other elements identified on the edge area, such as W (49.61 wt.%), Ti (21.86 wt.%), and N (12.03 wt.%) are components of insert material.

### 4.3. Machinability of HEA Considering Cutting Forces

The machining rating values are between 100% and 200% considering the cutting force components on axes. They characterize the HEA material when comparing with a reference material having similar properties (stainless-steel E 304).

During the cutting tests, the values of the three components of the cutting force are recorded, while, for the further processing, minimum, maximum and average values of the force calculated for a user-defined time period are supplied by the software. The diagrams in Figure 9a show slight differences between the two materials with regard to the values of the feed force *F_x_*, the linear evolution being almost overlapped (machining rating 100%). Following the absolute differences of the 25 measurement points, it has resulted that the three highest values of feed force were 19, 16 and 15 N.

The comparison diagrams of the cutting force (F_y_) are shown in Figure 9b and reveal significant differences. The cutting force diagram for HEA is situated above the similar diagram of stainless-steel. The linear evolution is quasi-parallel, the offset being of approximately 65 N. The maximum differences have appeared for the zones where the depth of cut has a maximum value (0.4 mm). The biggest difference of 130 N is registered for the Test 5. Another feature of the cutting force F_y_ of HEA is that its variation is much narrower, namely F_y HEA_ = −25 to −125 N, compared to that of the stainless-steel that has a variation of F_y 1810_ = −40 to −265 N. The machining rating for HEA on y-axis is 153%.

Figure 9c shows the axial force F_z_ evolution for the two materials. The diagram for HEA is situated beneath the diagram of the 304 stainless-steel (positive values) and it ranges between 50 and 190 N, the field being narrower than that of 50–300 N corresponding to the stainless-steel. The linear evolution is quasi-parallel, the offset being 55 N. The machinability rating of HEA with regard to the stainless-steel concerning the axial force F_z_ is 152%. It shows the better machinability behavior of HEA, being 52% better than that of the stainless-steel regarding this criterion. As a general conclusion for the assessment of machinability based on the F_y_ and F_z_ force values, it can be estimated that HEA is less influenced by the machining parameters than stainless-steel. The depth of cut a_p_ machining parameter has the most significant influence on the cutting force components, followed by the feed f_z_ and the cutting speed v_c_.

The machining rating values of 100%, 153%, and 152% are characterizing the HEA material on x, y and z axes, respectively. In addition, it becomes clear that they cannot offer a precise assessment of the machinability. For example, at a closer examination of the diagrams in Figure 9a,c, it is apparent that the force values corresponding to the processing of HEAs and stainless-steel are very close for Tests 2, 6, 10, 14, 18 and 22. For those surfaces, the common parameter is the depth of cut, having the smallest values for Levels 1 and 2. From this point of view, the machinability rating tends to be around 100%.

Considering the real cutting force given by Equation (1), the machining rating is variable in the range 117‒211% depending on the cutting parameters specific to each test (Figure 10). The trendline is almost horizontal, leading to an average of the machining rating for HEA of 159%. This value gives a closer estimation of the machinability of the studied HEA material. To give a quantitative assessment for the HEA material, in certain situations when necessary, it is possible to choose values of the cutting parameters in ranges larger by up to 59% than those recommended for stainless-steel. As usual, the maximum value should only be considered after some verification tests. It is recommended to use up to 75% of the machining rating increase, namely up to 45% increase of the cutting parameters.

### 4.4. Microstructure and Surface Changes during Machining

To estimate the changes that occur in the material during processing, both the surface of the chip and the processed surface were studied. Regarding the chips surface, the aspects regarding the deformation and shearing mode of the micro-volumes of material and the level of contamination or oxidation were highlighted (Figure 11). A higher plasticity of stainless-steel is evidenced by the formation of quasi-symmetrical deformation planes on scrap surfaces, while in the case of the HEA the breaking and sliding planes are irregular, due to the inhomogeneity of the metallic matrix.

In the case of HEA, frequent fragmentation and overlapping of micro volumes of material occurred. This aspect can be generated on the one hand by the inhomogeneity of the material, which was not initially subjected to a heat treatment of homogenization. On the other hand, it was found that the oxidation process is very intense on the surface of the chip; the oxide films and contaminants acquired from the working environment could act as fragmentation initiators.

Another explanation for the chip fragmentation may come from the microhardness analysis. Thus, the values measured in different areas of the HEA alloy ingot showed that the material is not homogeneous. The microhardness values fall within the range of 365–444 HV0.2, showing an average value of 414 HV, slightly greater than the reference value of 400 HV of the unprocessed material, and a small variation around the average between −11.85% and +7.3%. These values can be explained by the inhomogeneity of the cast material (difference in hardness between the phases of the alloy) or by self-hardening effect. As concerns the self-hardening effect, there is a correspondence between the maximum values of the cutting force component F_z_ (Tests 5, 9, 12, 17, 21 and 25) and the maximum values of the microhardness of the same tests.

The roughness of the machined surfaces in the cutting tests is in the range 0.39‒2.058 µm, and it is directly influenced by the feed and depth of cut. There are some areas that have roughness values affected by imperfections caused by material pulling out in the cutting process, as well as those caused by the imperfections of the alloy structure (cracks, craters and voids).

Analyzing the X-ray diffraction patterns it becomes obvious that the structure in the surface region, where the material reached the highest temperature during the processing, is similar to that of the bulk material. No new phases appeared after the process, the HEA material withstanding very well the heating cycles. The multitude of phases identified on the machined area, having different hardness and plasticity, can be an important factor on the wear amplitude measured on cutting edge. Some of these intermetallic phases (Fe_2_Al_5_ and FeAl_3_) have elastic moduli higher than the modulus of pure Fe (about 200 GPa) [42]. The hardness of Al-rich phases depends on their size and crystallographic orientation, chemical composition and applied load [43]. During cutting, when the sharp edge of the tool press and excavate the inhomogeneous material, some of hard phases act as a sharp micro-edge contributing to exfoliation of protective layer of tool surface.

The performed studies represent a new approach to HEA materials in terms of machinability and also preliminary research that opens a new path. The article supplies a comprehensive image of the HEA material behavior during milling, in comparison with 304 stainless-steel trademark. Recommendations on cutting parameters to be used in industrial applications are given. More and detailed research should be achieved in order to have a more precise image of the machinability of the HEA by considering the cutting speed, material removal rate and cutting tool life.

## 5. Conclusions

The performed studies represent a new approach to HEA materials in terms of machinability and preliminary research that opens a new path. The article supplies a comprehensive image of the HEA material behavior during milling, in comparison with SAE 304 stainless-steel. Therefore, the present work evaluates seven parameters influencing HEA machinability compared to papers dedicated to this type of material, where maximum two are discussed. In addition, it analyzes more parameters than many articles that study the machinability of different types of alloys. Recommendations on cutting parameters to be used in industrial applications are given that are of great importance for the application of machined HEAs in a large pallet of domains outlined in the article.

Preliminary cutting tests showed that conventional cutting tool materials, such as rapid steel, have very limited life and suffer an abrasion wear process resulting in a large amount of heat in the cutting process. They cannot be used for HEA cutting. The cutting tests proved that inserts used for high feed milling having carbide grade HSC 05 and coating PVTi can be used in good conditions for HEA machining without coolant. Insert wear values measured after cutting tests corresponding to a machining time tm = 10 min show maximum values on the rack face KB max = 490 µm and KM max = 225 µm; for the larger crater, KB = 134 µm and KM = 83 µm; and, on the flank, VBB max = 67 µm, which indicate that the insert wear still belongs to the steady-state wear region.

In terms of machinability, the research shows that, taking into account the real cutting force, the machining value of HEA is variable in the range 117‒211% depending on the cutting parameters and compared to stainless-steel. Therefore, the behavior of the HEA is found to be about 59% better than that of the stainless-steel. This value may be used for choosing the cutting parameters for HEA with 45% greater than those for the stainless-steel (usual recommendation), going up to 59% in certain situations when necessary.

The chemical analyses performed on chip surface indicate a high oxidation tendency for both 304 stainless-steel and HEA. In terms of microstructure phase’s evolution during machining, the XRD analysis demonstrates that the patterns are very similar for as-cast and machined surfaces. During cutting process, changes occur in the material microstructure, even for low temperature values (251 °C). XRD analysis performed at different angles of inclination in relation to the examined surface showed that, in addition to the two main phases identified in the as-cast alloy, series of complex compounds are also formed.

The measured values of microhardness (365–444 HV0.2) can be explained by the inhomogeneity of the cast material (difference in hardness between the phases of the alloy) or by self-hardening effect caused mainly by the cutting force component F_z_. The microhardness is proved to be one cause of the chip fragmentation mode.

More detailed research should be achieved to have a more precise image of the machinability of the HEA by considering the cutting speed, material removal rate and cutting tool life. Furthermore, extended ranges of the cutting parameters should be considered for both finish cut and rough cut and to generate predictive mathematical models to improve processing and optimize cutting parameters.

The performed studies represent a new approach to HEA materials in terms of machinability and also preliminary research that opens a new path. Preliminary cutting tests showed that conventional cutting tool materials, such as rapid steel, have very limited life and suffer an abrasion wear process resulting in a large amount of heat in the cutting process. They cannot be used for HEA cutting. The cutting tests proved that inserts used for high feed milling having carbide grade HSC 05 and coating PVTi can be used in good conditions for HEA machining without coolant.

In terms of machinability, when taking in consideration the real cutting force, the behavior of the HEA is found to be about 59% better than that of the stainless-steel. This value may be used for choosing the cutting parameters for HEA with 45% greater than those for the stainless-steel (usual recommendation), going up to 59% in certain situations when necessary.

## Figures and Tables

**Figure 1 materials-13-04181-f001:**
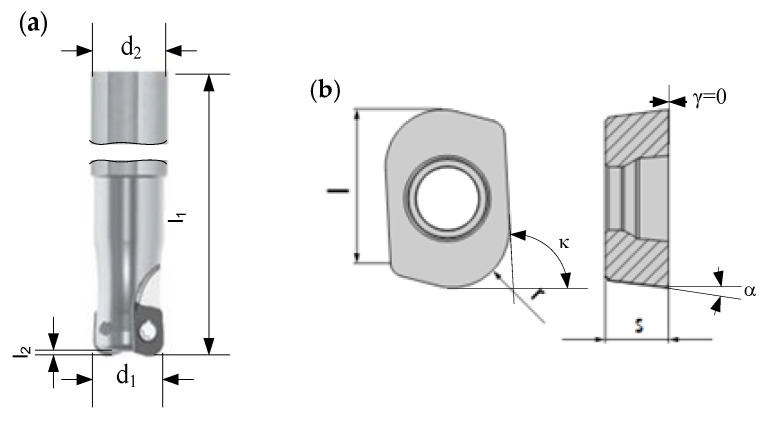
Milling cutter used for cutting tests: (**a**) milling cutter Slotworx HP Pokolm; and (**b**) insert 02 66 860 R20 and milling cutter body 3 36 12 166 G’4.

**Figure 2 materials-13-04181-f002:**
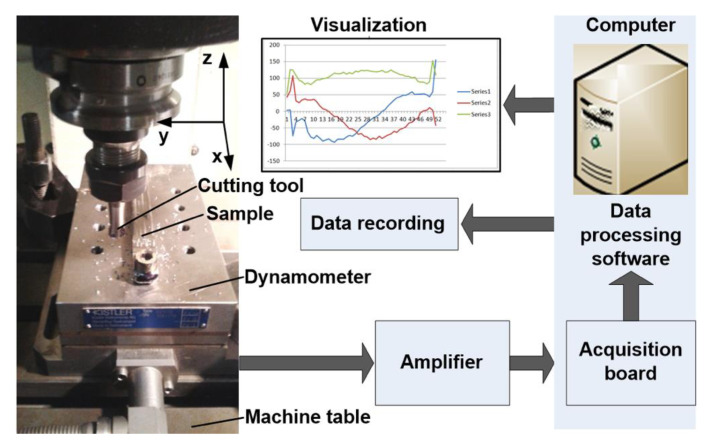
Experimental setup used in measuring of the cutting force components in milling of HEA and stainless steel.

**Figure 3 materials-13-04181-f003:**
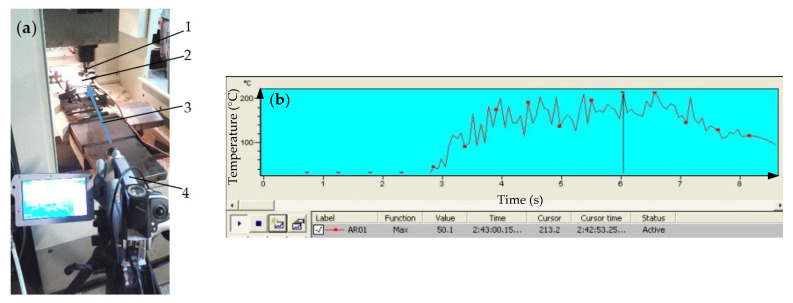
Experimental setup for measuring temperature during machining: (**a**) equipment used for measuring temperature in cutting area (1, cutting tool; 2, sample; 3, camera side direction of viewing of the cutting area; and 4, thermo vision camera); and (**b**) time-temperature evolution during cutting.

**Figure 4 materials-13-04181-f004:**
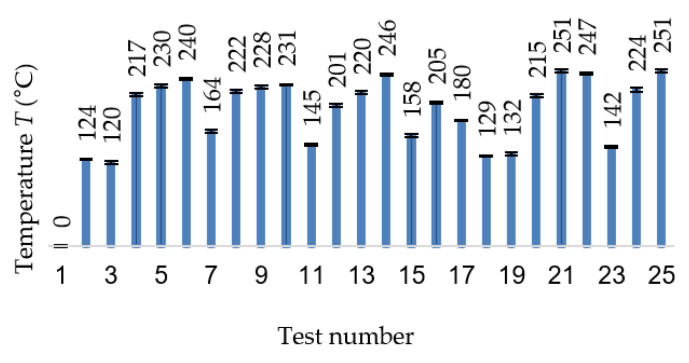
Measured temperature during cutting tests.

**Figure 5 materials-13-04181-f005:**
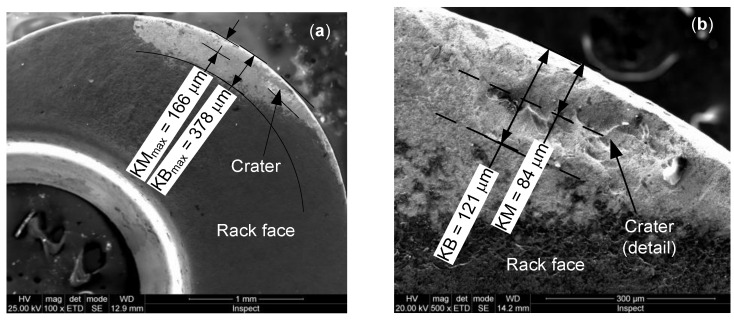
SEM images of wear on Cutting Edge 1 of the insert after cutting of HEA: (**a**) wear on rack face; (**b**) detail of the crater on the rack face; (**c**) wear on flank; and (**d**) detail of the worn flank.

**Figure 6 materials-13-04181-f006:**
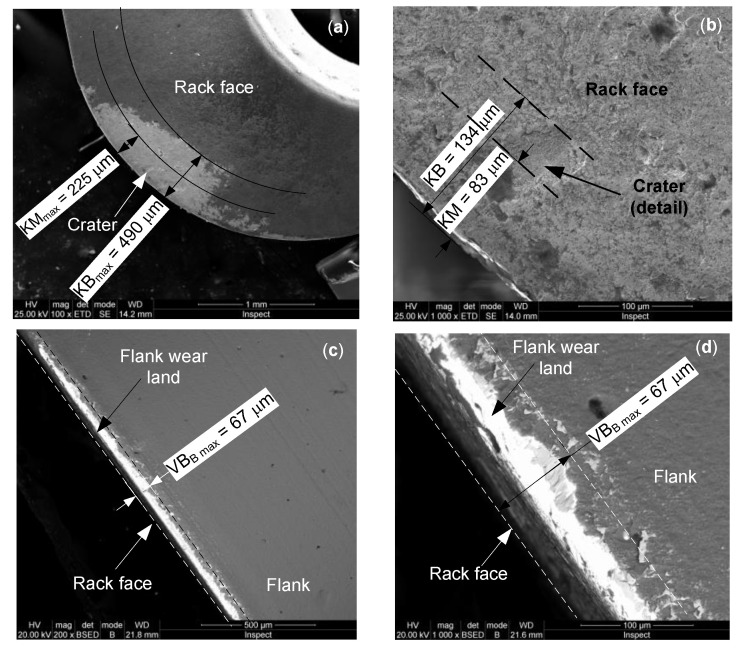
SEM images of wear on Cutting Edge 2 of the insert after cutting of HEA: (**a**) wear on the rack face; (**b**) detail of the crater on the rack face; (**c**) wear on the flank; and (**d**) detail of the worn flank.

**Figure 7 materials-13-04181-f007:**
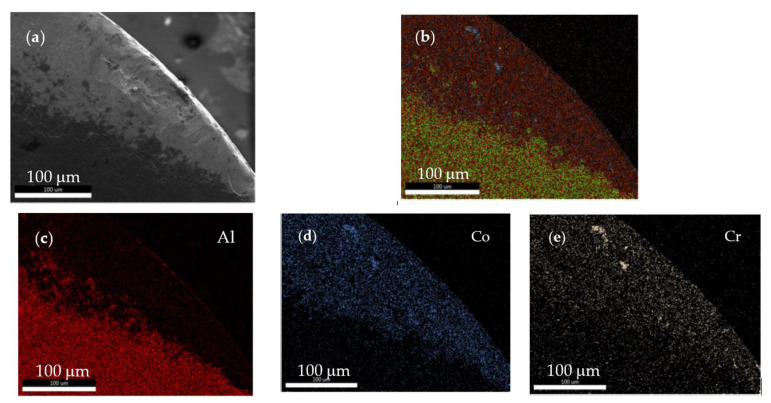
Distribution of chemical elements on the wear zone of cutting edge: (**a**) SEM image of cutting edge; (**b**) global aspect of chemical elements distribution on cutting edge; (**c**) Al distribution; (**d**) Co distribution; and (**e**) Cr distribution.

**Figure 8 materials-13-04181-f008:**
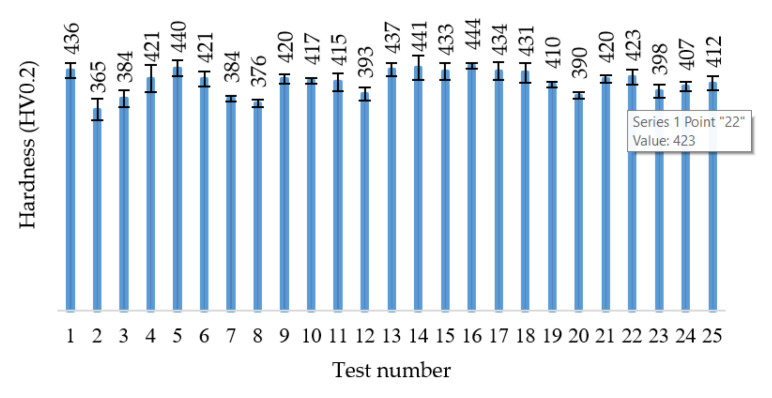
Hardness variations after machining using different cutting parameters.

**Figure 9 materials-13-04181-f009:**
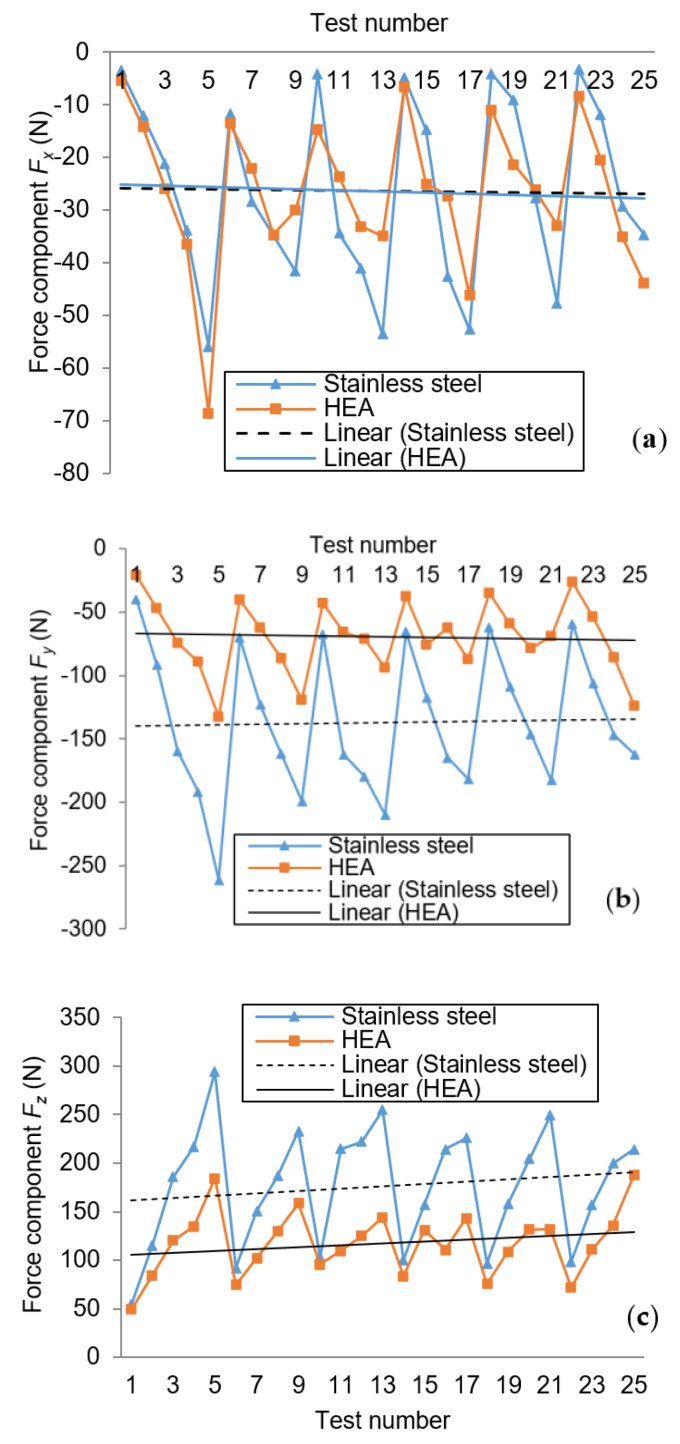
Comparison chart of cutting force components for HEA and stainless-steel: (**a**) evolution of F_x_; (**b**) evolution of F_y_; and (**c**) evolution of F_z_.

**Figure 10 materials-13-04181-f010:**
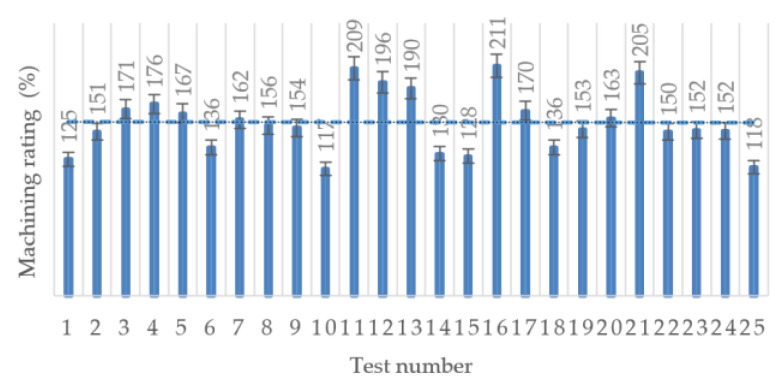
Machining rating in case of HEA with regard to the stainless-steel E 304.

**Figure 11 materials-13-04181-f011:**
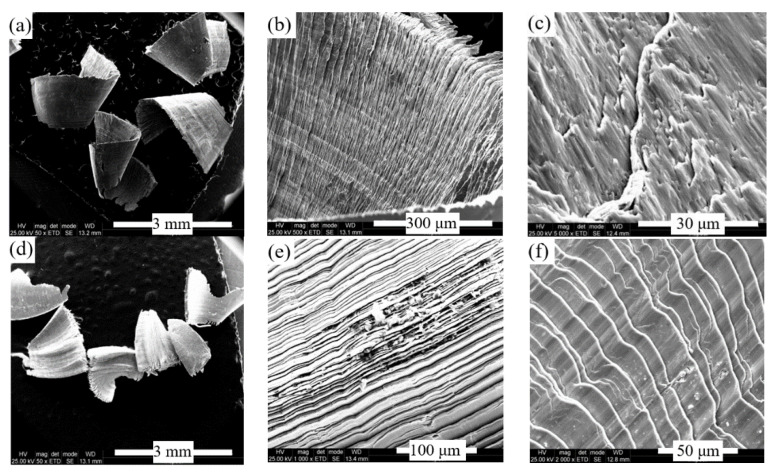
Aspect of chips resulting from the cutting process in Experiment 13: (**a**) fragments of HEA chip; (**b**) the internal aspect of the chip that highlights the sliding and fragmentation planes in HEA; (**c**) appearance of the external surface of the chip with sliding on different planes in HEA; (**d**) fragments of chip for 304 stainless-steel; (**e**) strips on the internal surface of the chip and small accumulations of impurities (304); and (**f**) striped appearance of the external surface of the chip (304).

**Figure 12 materials-13-04181-f012:**
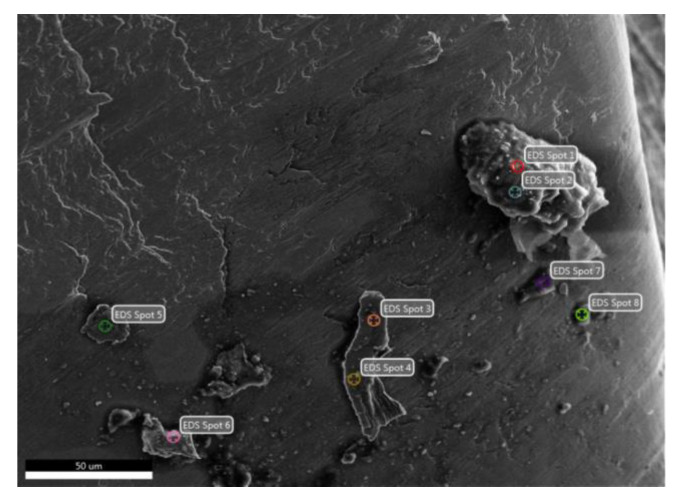
Chip surface of HEA with adherent micro-zones resulting from the cutting process in Experiment 13.

**Figure 13 materials-13-04181-f013:**
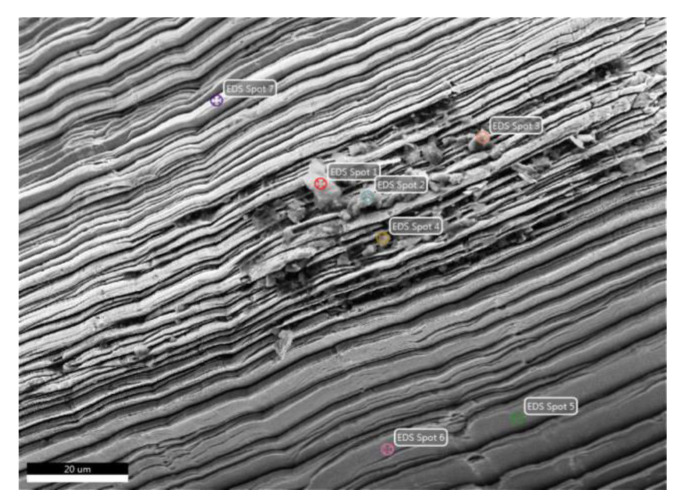
Chip surface of 306L and adherent micro-zones resulting from the cutting process in Experiment 13. Spots 1–7 are the selected micro-zones for chemical composition analysis.

**Figure 14 materials-13-04181-f014:**
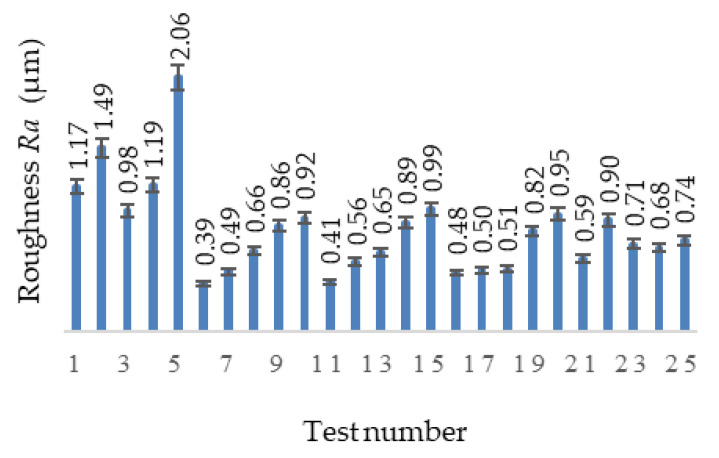
Surfaces roughness during the tests.

**Figure 15 materials-13-04181-f015:**
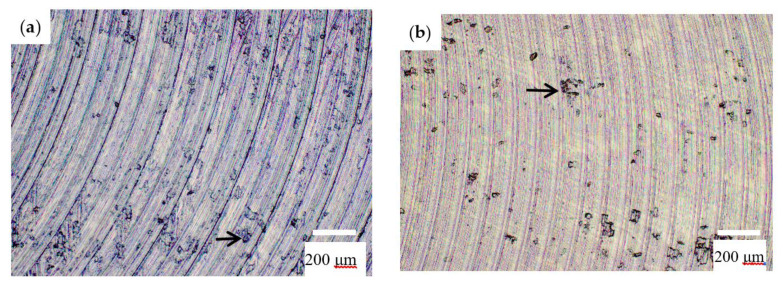
Quality of machined surface for HEA: (**a**) aspect of machined surface in Test 4, v_c_ = 20 m/min, f_z_ = 0.1 mm/tooth and a_p_ = 0.34 mm; and (**b**) aspect of machined surface in Test 8, v_c_ = 25 m/min, f_z_ = 0.09 mm/tooth and a_p_ = 0.34 mm. Arrows indicate the craters resulted on the machined surface.

**Figure 16 materials-13-04181-f016:**
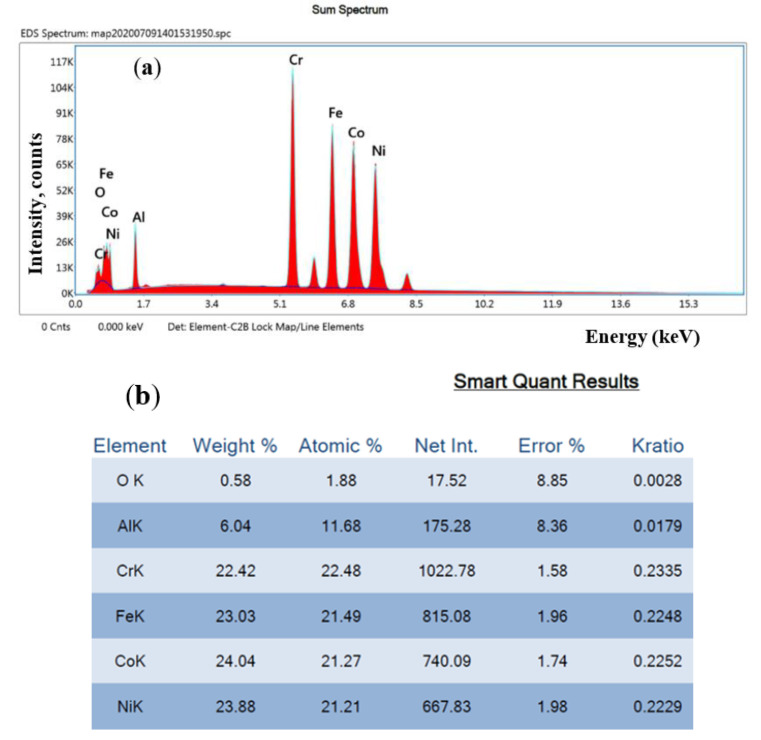
Chemical composition measurements on machined surface of Test 25: (**a**) chemical element spectrum; and (**b**) chemical element concentration.

**Figure 17 materials-13-04181-f017:**
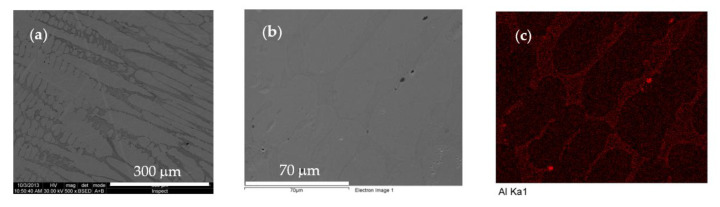
Microstructure of as-cast Al_0.6_CoCrFeNi alloy: (**a**) aspect on cross section; (**b**) micro-area selected for chemical analysis; and (**c**) aluminum distribution in selected area. Sum spectrum wt.%: Al (6.87); Cr (20.66); Fe (24.18); Co (24.04); and Ni (24.26).

**Figure 18 materials-13-04181-f018:**
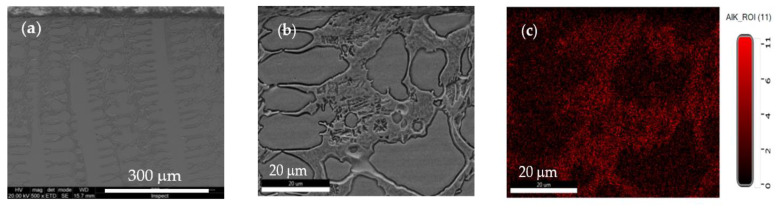
Microstructure of machined Surface 25 of Al_0.6_CoCrFeNi alloy: (**a**) aspect on cross section; (**b**) micro-area selected near the machined surface; and (**c**) aluminum distribution in the selected area. Sum spectrum wt.%: Al (9.90); Cr (16.99); Fe (20.02); Co (23.40); and Ni (29.70).

**Figure 19 materials-13-04181-f019:**
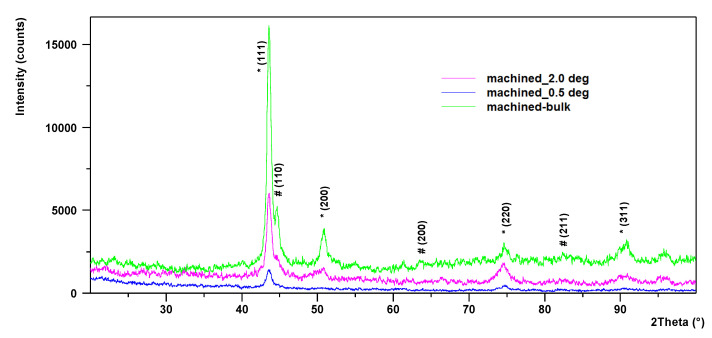
X-ray diffraction patterns acquired from the machined sample at different incidence angle.

**Table 1 materials-13-04181-t001:** Chemical composition of experimental Al_0.6_CoCrFeNi alloy and stainless steel 304 obtained through spectrometric tests.

Al_0.6_CoCrFeNi		**Chemical Elements**						
		**Al**	**Cr**	**Fe**	**Co**	**Ni**	**Mn**	**Mo**
	Weight (%)	6.68	21.47	23.12	24.36	24.36	-	-
	Atomic (%)	13.04	21.74	21.74	21.74	21.74	-	-
Stainless steel 304	Weight (%)	0.1	18.6	~70	-	9.44	1.17	0.41

**Table 2 materials-13-04181-t002:** Levels of parameters for experimental plan during milling HEA.

Parameter	Symbol (Units)	Levels				
		1	2	3	4	5
Cutting speed	v_c_ (m/min)	20	25	30	35	40
Feed	f_z_ (mm/tooth)	0.05	0.07	0.09	0.1	0.15
Axial depth of cut	a_p_ (mm)	0.1	0.18	0.26	0.34	0.42

**Table 3 materials-13-04181-t003:** General experimental plan according to Taguchi method used for cutting tests.

Test	Cutting Speed v_c_ (m/min)	Feed f_z_ (mm/tooth)	Axial Depth of Cut a_p_ (mm)
1	1	1	1
2	1	2	2
3	1	3	3
4	1	4	4
5	1	5	5
6	2	1	2
7	2	2	3
8	2	3	4
9	2	4	5
10	2	5	1
11	3	1	3
12	3	2	4
13	3	3	5
14	3	4	1
15	3	5	2
16	4	1	4
17	4	2	5
18	4	3	1
19	4	4	2
20	4	5	3
21	5	1	5
22	5	2	1
23	5	3	2
24	5	4	3
25	5	5	4

**Table 4 materials-13-04181-t004:** The chemical composition on adherent micro-zones of HEA chips, wt.%.

Fe	Cr	Ni	O	Ca	Si	Al	Co	C	Measurement Area
4.18	2.71	1.44	69.92	13.47	-	4.62	2.32	-	1
0.81	0.61	0.4	54.25	-	-	0.72	0.52	42.69	2
92.06	2.52	0.83	1.06	-	-	0.51	3.02	-	3
93.43	2.36	0.68	0.09	-	-	0.55	2.89	-	4
80.54	2.8	0.85	3.93	-	-	8.76	3.11	-	5
52.28	3.29	1.03	-	-	3	1.09	-	36.89	6
22.77	15.67	9.82	6.34	9.68	11.16	6.44	-	-	7
8.37	3.65	2.48	16.3	-	-	1.51	2.94	61.73	8

**Table 5 materials-13-04181-t005:** The values of the local chemical composition for the micro-zone analyzed in Figure 10e, wt.%.

Fe	Cr	Ni	Mo	O	Ca	Si	Al	Measurement Area
17.64	5.21	2.92	13.42	60.78	-	-	-	1
8.20	1.79	0.95	-	54.57	29.66	4.83	-	2
5.68	1.93	0.98	-	48.03	-	38.01	5.37	3
16.88	-	-	2.62	43.92	24.01	9.46	3.11	4
67.68	25.17	7.15	-	-	-	-	-	5
54.93	35.35	9.72	-	-	-	-	-	6
51.17	34.88	13.95						7

**Table 6 materials-13-04181-t006:** Phase composition of the HEA alloy.

No.	Ref. Code	Chemical Formula	Score	SemiQuant (%)
1	04-022-2301	Cr0.25 Fe0.25 Co0.25 Ni0.25	40	19
2	04-012-3422	Cr0.6 Fe0.4 Co2 Al	39	6
3	04-006-3945	Fe Al	36	13
4	04-003-2931	Ni1.2 Al0.8	28	23
5	04-018-8010	Co Ni2 Al4	35	8
6	04-018-6762	Co0.74 Ni0.7 Al0.56	24	1
7	04-009-2288	Co2 Ni Al9	35	30

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
