# Peer review of "Cutting Behavior of Al0.6CoCrFeNi High Entropy Alloy"

_materials, 2020, doi:10.3390/ma13184181_

Round 1

Reviewer 1 Report

Lines 426-427: The large percentages of chemical elements, such as C (42 wt.% C), Ca (13 wt.%) and Si (11 wt.%) can be due to impurities from the cooling environment.

Suggestion: Adherent deposits on the chips probably come from the working environment (tools or handling by operators). Cooling environment was air which not contains Si, Ca and C!

Author Response

Thank you very much for suggestions.

The sentence from - lines 426-427 has been modified according your suggestion.

Reviewer 2 Report

All comments were properly replied and the paper has been well revised.

The modifications have contributed to substantially improve the quality of the research and I consider that the paper is suitable for publication.

Author Response

Thank you very much for the apreciations and suggestions that contribute to improvement of the paper content.

Reviewer 3 Report

The manuscript entitled: 'Cutting Behavior of Al0.6CoCrFeNi High Entropy Alloy' deals with the cutting behavior of the HEA alloy. I have the following concerns with the manuscript.

  • Fig. 3 (b and c) - legends and their units on the X and Y axis are missing.
  • The scale on the image in Tables 4 and 5 are not-readable. Also, the text in the figure in Tables 4 and 5 are not visible/readable.
  • The features observed in Figure 13 should be directly marked in the images.
  • Fig. 14(a) Y-axis legend/unit is missing.
  • Fig. 17 and 18 do not give much information. Maybe clubbed with Fig. 16.
  • The peaks should be identified and should be marked directly in the XRD pattern.
  • Fig. 20 does not add much information to the manuscript.
  • Conclusions should be precise and concise. Presently, the conclusion is vague and needs to be reduced to highlight the important aspects of this manuscript.
  • A strong scientific discussion is missing!
  • Typos in the manuscript have to be rectified carefully.

Author Response

Answers for Reviewer 3

First of all, the authors would like to thank you for your suggestions and comments.

We consider that the paper became more interesting and better written after we made these corrections.

  1. 3 (b and c) - legends and their units on the X and Y axis are missing.

Ans. The legend and their units have been written. We eliminated the figure 3c, as irrelevant.

  1. The scale on the image in Tables 4 and 5 are not-readable. Also, the text in the figure in Tables 4 and 5 are not visible/readable.

 Ans. We rearranged the figures and enlarged them to be visible.

  1. The features observed in Figure 13 should be directly marked in the images.

Ans. We used arrows for indicate the imperfections on the Figure 15 (former Figure 13).

  1. Fig. 14(a) Y-axis legend/unit is missing.

Ans. We inserted Y-axis legend and units in Figure 16 (former Figure 14).

  1. Fig. 17 and 18 do not give much information. Maybe clubbed with Fig. 16. The peaks should be identified and should be marked directly in the XRD pattern.

Ans. We eliminate former Figures 17 and 18 . We inserted the new Figure 19, containing all suggested data.

  1. Fig. 20 does not add much information to the manuscript.

Ans. We eliminated Figure 20.

  1. Conclusions should be precise and concise. Presently, the conclusion is vague and needs to be reduced to highlight the important aspects of this manuscript.

Ans. We rewritten the conclusions in adequate manner.

  1. A strong scientific discussion is missing!

Ans.We created a new Chapter (no. 4) dedicated to Discussions!

  1. Typos in the manuscript have to be rectified carefully.

Ans. Typos have been rectified in all manuscript.

Reviewer 4 Report

The paper presents a detailed study of the cutting behavior of the Al0.6CoCrFeNi high entropy alloy, and found the cutting properties are much better than that of stainless steels. X-ray diffraction investigations showed no changes induced by cutting in the structure of the sample. The wear effects that appear on the cutting edge faces for the tool made of rapid steel compared to carbide during HEA machining lead to the conclusion that
Physical Vapor Deposition (PVD)-coated carbide inserts are proved to be suitable for the cutting of HEAs. The paper was very nice and can be accepted as is.

Author Response

The authors would like to thank you very much for the suggestions and appreciations regarding our paper.

Round 2

Reviewer 3 Report

The authors have satisfactorily addressed the comments. However, there is still room for improvement. 

Nevertheless, I may reluctantly recommend the manuscript for publication in the present state.

Author Response

Thanks very much for your comments

This manuscript is a resubmission of an earlier submission. The following is a list of the peer review reports and author responses from that submission.

Round 1

Reviewer 1 Report

The manuscript entitled: 'Cutting Behavior of AlCoCrFeNi High Entropy Alloy' focuses on the cutting characteristics of the high entropy alloys. The concerns are:

  • The manuscript should include the phases (XRD) and microstructure (SEM) of the HEA alloy before and after the machining test. Without the phase and microstructural information, there is no point in discussing the mechanism.
  • A strong scientific discussion is missing correlating the wear mechanism with the phases present.
  • Is there any oxidation of the HEA after machining?
  • Error bars should be introduced for all data points.
  • Typos in the manuscript should be rectified carefully. For instance, space should be given between number and unit.

Reviewer 2 Report

The objective of the work is to evaluate the machinability of AlCoCrFeNi High Entropy Alloy but the analysis carried out is not deep enough and the conclusions are weakly presented and are not well documented. Comparison with 304 stainless steel is not made for all studies and information is missing for this material. For example, the study of cutting temperature is missing for 304 stainless steel. A surface roughness study for this material has also not been performed. It is necessary to improve the analysis of tool wear. Figure 5 should be improved (wear regions are not shown correctly). Furthermore, tool wear study has not been performed for the 25 tests. In section 3.2 (268-269) the authors say “For the cutting tests made on stainless steel, the inserts has been rotated by 180° to starting the processing with new cutting edges” but the results of tool wear are not presented. The microhardness study does not allow to draw relevant conclusions about machinability.

I recommend adding a table with the conditions of the 25 tests carried out for a better interpretation of the results.

Paper redaction must be reviewed and English writing should be improved.

Reviewer 3 Report

Dear Authors,

I have read your very well written manuscript carefully and I would say that this manuscript would be very interesting for readers because it concerns the cutting behavior of high entropy alloy (AlCoCrFeNi) and the area of this research work is important form the HEA development point of view. The objectives of the study are clearly defined. The introduction provides a good, generalized background of the topic. The results are clearly explained and are presented in an appropriate format. The figures show essential data; some of the data are also summarized in the text. The cited literature is relevant to the study and balanced. I recommend to publish this manuscript.